# Genome-Wide Identification and Characterization of Lectin Receptor-Like Kinase Gene Family in Cucumber and Expression Profiling Analysis under Different Treatments

**DOI:** 10.3390/genes11091032

**Published:** 2020-09-02

**Authors:** Duo Lv, Gang Wang, Liang-Rong Xiong, Jing-Xian Sun, Yue Chen, Chun-Li Guo, Yao Yu, Huan-Le He, Run Cai, Jun-Song Pan

**Affiliations:** School of Agriculture and Biology, Shanghai Jiao Tong University, Shanghai 200240, China; lvduocloudy@sjtu.edu.cn (D.L.); wg@shu.edu.cn (G.W.); xlr1394981178@sjtu.edu.cn (L.-R.X.); leyunu@sjtu.edu.cn (J.-X.S.); yuechen321@sjtu.edu.cn (Y.C.); jackling@sjtu.edu.cn (C.-L.G.); yuyaosjtu@sjtu.edu.cn (Y.Y.); hlhe75@sjtu.edu.cn (H.-L.H.); cairun@sjtu.edu.cn (R.C.)

**Keywords:** cucumber, expression analysis, genome-wide analysis, lectin receptor-like kinase, phylogenetic analysis

## Abstract

Lectin receptor-like kinases (LecRLKs) are a class of membrane proteins found in plants that are involved in diverse functions, including plant development and stress responses. Although *LecRLK* families have been identified in a variety of plants, a comprehensive analysis has not yet been undertaken in cucumber (*Cucumis sativus* L.). In this study, 46 putative *LecRLK* genes were identified in the cucumber genome, including 23 G-type and 22 L-type, and one C-type *LecRLK* gene. They were unequally distributed on all seven chromosomes, with a clustering tendency. Most of the genes in the cucumber *LecRLK* (Cs*LecRLK)* gene family lacked introns. In addition, there were many regulatory elements associated with phytohormones and stress on these genes’ promoters. Transcriptome data demonstrated distinct expression patterns of *CsLecRLK* genes in various tissues. Furthermore, we found that each member of the *CsLecRLK* family had its own unique expression pattern under hormone and stress treatment by the quantitative real-time PCR (qRT-PCR) analysis. This study provides a better understanding of the character and function of the *LecRLK* gene family in cucumber and opens up the possibility to exploring the roles that *LecRLK*s might play in the life cycle of cucumber.

## 1. Introduction

In order to better adapt to the living environment, plants have evolved a complete set of signal receptor proteins over the course of their evolution. After receiving external stimulus, they transmit signals to downstream pathways to allow plants to respond to the stimulus. Cell-surface receptors, a kind of signal receptor proteins, play important roles in receiving and transmitting environmental signals. The receptor-like kinase (RLK) family, one important family of cell surface receptors, contains three domains, such as the extracellular domain, transmembrane domain (TM) and intracellular kinase domain [1]. RLK proteins could be classified into different families based on the structure of the extracellular domains [2].

Lectin receptor-like kinases (LecRLKs), a class of RLKs that contain a lectin domain within the extracellular domain [1], are a gene family that is specialized for sensing external environmental stimuli and transmitting signals. They are localized on the cell membrane, relying on N terminus diverse extracellular ligand recognition domains (also called lectin domain) to recognize various environmental stimuli, and then phosphorylate downstream proteins through their C terminus intracellular kinase domain to pass on received signals [1,3].

Based on the identity of the lectin domains, the LecRLKs have been divided into three subfamilies (Figure 1): L-type, G-type and C-type LecRLKs [4]. These subclasses are very distinct from each other, differentiated by the sugar-binding ability of the lectin domain. The G-type LecRLKs were previously named the B-type LecRLKs, due to the resemblance of their extracellular domains to bulb lectin proteins [5]. G-type LecRLKs were also historically known as S-domain RLKs, because apart from containing bulb lectin protein, its extracellular domain sometimes contains an S-locus region, known for its role in self-incompatibility reactions [5,6]. In most proteins (but not necessarily in all), they are also accompanied by an Epidermal Growth Factor (EGF) motif and/or a Plasminogen Apple Nematode (PAN) motif [4] (Figure 1B). The EGF motif contains some cysteines and is involved in the formation of disulfide bonds [4,7]. The PAN motif is believed to be related to protein–protein and protein–carbohydrate interactions [8]. As the name suggests, the lectin domain of L-type LecRLKs resembles the soluble lectin protein found in leguminous plants [9]. The third class of LecRLKs is C-type lectin kinases. The C-type LecRLKs in plants are thought to be homologues of calcium-dependent lectin motifs, which are a large group of mammalian proteins known to be involved in innate immune responses and pathogen recognition [10]. Although C-type lectins are present in large numbers in mammalian systems, C-type LecRLKs seem to be scarce in plants [3]. For example, only one gene encoding for C-type LecRLK exists in both rice and *Arabidopsis* [4].

At present, the existing research on the biological functions of LecRLKs is not very extensive [5]. Some studies suggest that LecRLKs are mainly involved in plant development, stress response and innate immune responses. For example, LecRK-b2, an L-type receptor-like kinase in *Arabidopsis*, can be induced by abscisic acid, salt and osmotic stress [11]. The promoter of *LecRK-b2* can be activated by transcription factor ABSCISIC ACID INSENSITIVE3, which mediates abscisic acid (ABA) responses in seeds [11]. Another L-type receptor-like kinase, LECRK-IV.2, plays an important role in the sterility of *Arabidopsis*. In the mutant of LECRK-IV.2, all pollen grains are deformed and collapsed [12]. In rice, OslecRK can maintain seed viability by regulating the expression of α-amylase genes. The mutant of OslecRK reduces the resistance of rice plants to fungal and bacterial pathogens as well as herbivorous insects [13].

Genome-wide analysis of the *LecRLK* gene family has been reported for some plants [4,14,15,16], but there are only a few early studies reporting on the presence of *LecRLK*s in Cucurbitaceae crops [17]. A comprehensive understanding of *LecRLK* genes in cucumber is still lacking. In this study, we used bioinformatics methods to identify *LecRLK* genes from the cucumber genome and analyzed their phylogenetic relationships, gene structure, conserved domain, gene duplications, chromosome distribution and *cis*-acting elements on the promoters. Finally, we profiled the expression of the predicted genes in different tissues and response to gibberellin (GA), abscisic acid (ABA), 1-Naphthaleneacetic acid (NAA), indole-3-acetic acid (IAA) and cold treatments in cucumber. Our study provides valuable information for further functional research on the *LecRLK* gene family in cucumber.

## 2. Materials and Methods

### 2.1. Identification of LecRLK Genes in Cucumber

The whole genome and protein sequence data of cucumber were downloaded from a public database (http://cucurbitgenomics.org/; Cucumber (Chinese Long) genome v2). The Hidden Markov Model (HMM) was used to identify cucumber *LecRLK* candidates, and the HMM profiles of LecRLKs were downloaded from the Pfam protein database (http://pfam.xfam.org/). The models for these files were L-type (Lectin_legB PF00139), G-type (B_lectin PF01453) and C-type (Lectin_C PF00059). We used HMMER 3.0 [18] to search three types of genes from cucumber protein sequence data with e-value cutoffs of <0.001. We then submitted protein sequences of these genes to two bioinformatics websites, Pfam [19] (http://pfam.xfam.org/) and SMART (2019) [20] (http://smart.embl.de/), with an e-value cut-off of 1.0, while retaining those genes that contained lectin domain, transmembrane domains and kinase domains at the same time. The *LecRLK* family data of rice and *Arabidopsis* were taken from a previous research literature [4].

### 2.2. Phylogenetic Analysis

The full-length protein sequences of CsLecRLKs were aligned using the MUSCLE program (http://www.ebi.ac.uk/Tools/msa/muscle/) with the default parameters [21]. The phylogenetic tree was constructed through the Neighbor-Joining (NJ) method using MEGAX_software v.10.1.8 with the following parameters: Poisson model, pairwise deletion and 1000 bootstrap replications. The phylogram was rooted using two distantly related receptor kinase (RK) sequences of cucumber, CsaLRR-RK (Csa3G126190) and CsaPERK1 (Csa2G004170). We also chose At1G52310 (C-type LecRLK of *Arabidopsis*), At4G04960 (L-type LecRLK of *Arabidopsis*) and At1G61550 (G-type LecRLK of *Arabidopsis*) as outgroups of each LecRLK type.

### 2.3. Conserved Domain, Motif Identification and Gene Structure Analysis

The conserved motifs of the CsLecRLKs were predicted using the MEME program v.5.1.1 [22] (http://meme-suite.org/). The parameters were set as any number of repetitions, an optimum motif width of 6–210 residues, and 10 motifs were searched for. The site distribution mode was “zoops”, and the maximum search time was 18,000 s. The CDD program (https://www.ncbi.nlm.nih.gov/cdd/) was used to predict the function of each conservative motif with an e-value threshold of 0.01. Gene Structure Display Server [23] (http://gsds.cbi.pku.edu.cn/) was used to show the exon–intron structures of *CsLecRLK* genes. Adobe Illustrator CS6 was used to enhance figures.

### 2.4. Gene Location and Duplication Analysis of CsLecRLKs

A series of in-house Perl scripts was used to retrieve the location information for each *CsLecRLK* and the length of each cucumber chromosome from cucumber whole-genome sequence data, which were downloaded from a public database (http://cucurbitgenomics.org/) (Cucumber (Chinese Long) genome v2). Location information of each *CsLecRLK* and length of each chromosome were then displayed as visualized pictures through MapChart [24]. The Adobe Illustrator CS6 was used to enhance the picture.

We used two methods to find duplication events among the *CsLecRLK*s. One was confirming gene duplication with three criteria using in-house Perl scripts: (a) the shorter aligned gene covered >70% of the longer gene in length, (b) the similarity of aligned sequences was >70% [25,26] and (c) two genes located in the same chromosomal fragment of less than 100 kb and separated by five or fewer genes were identified as tandem duplicated genes [27]. Another was using the Multiple Collinearity Scan toolkit (MCScanX) to analyze the gene duplication events, with the default parameters [28].

Ks (synonymous substitution rate) and Ka (nonsynonymous substitution rate) values of tandem duplicated genes were calculated by the method of Nei and Gojobori as implemented in KaKs_calculator [29] based on the coding sequence alignments. The divergence time was calculated based on the formula T = Ks/2r, with Ks being the synonymous substitutions per site and r being the rate of divergence for nuclear genes from plants. r was taken to be 1.5 × 10^−8^ synonymous substitutions per site per year for dicotyledonous plants [30].

### 2.5. Analysis of Cis-Acting Elements

The upstream 1500 bp of each *CsLecRLK* was obtained from the genome annotation files of Cucurbit Genomics Databases (http://cucurbitgenomics.org/) via a series of in-house Perl scripts and the *cis*-acting elements contained in these sequences were then scanned using the Plantcare Databases (http://bioinformatics.psb.ugent.be/webtools/plantcare/html/). In the process of analysis, we filtered out the *cis*-acting elements that are ubiquitous in most genes, for example CAAT-box, TATA-box, TATC-box and so on, only showing those that may be typical and functional *cis*-acting elements presented in the *CsLecRLK* gene family. The structures and annotations of the promoters were then generated using GSDS v.2.0 (http://gsds.cbi.pku.edu.cn/) [28].

### 2.6. Expression Pattern Analysis

To analyze the expression profiles of cucumber *CsLecRLK* genes in different organs, we retrieved public RNA-seq data (accession number: SRP071224) from the National Center for Biotechnology Information Short Read Archive database. Analysis of RNA-seq data from 10 sampled cucumber tissues was performed based on the regular protocol by Wei et al. [31]. These cucumber tissues included root (4 week old seedlings), hypocotyl (4 week old seedlings), cotyledon (4 week old seedlings), true leaf (4 week old seedlings), stem (12 week old cucumber plants), tendril (12 week old cucumber plants), female flower (12 week old cucumber plants), male flower (12 week old cucumber plants), ovary (unfertilized, 12 week old cucumber plants) and peel (unfertilized ovary, 12-week-old cucumber plants). The Heatmap of the gene Log2 (FPKM + 1) values in 10 tissues of *CsLecRLK*s was drawn by the R program (3.5.2). To show which genes are widely expressed in different tissues, we also selected five typical tissues, including root (4 week old seedlings), hypocotyl (4 week old seedlings), cotyledon (4 week old seedlings), true leaf (4 week old seedlings) and tendril, to draw Venn diagrams using the R program (3.5.2).

### 2.7. Hormones and Cold Treatments

The typical cucumber line “9930” was used as the experimental material to investigate the expression pattern in response to various phytohormones and cold treatments. Cucumber seeds were soaked in 55 °C water for 2 h and germinated on a petri dish in a growth chamber at 28 °C in the dark for 2 days. The germinated seeds were grown in pots containing a peat/vermiculite mixture (3:1) in the greenhouse of Shanghai Jiao Tong University, and the controlled environment growth chamber was programed for light 16 h/25 °C and dark 8 h/20 °C. After germination for 4 weeks, the seedlings were placed into hydroponic boxes with 1/2 Murashige & Skoog (MS) liquid solution (pH 5.8, without sugar) for 1 week to adapt to the environment (the root was lucifugal), and were then treated with 100 mM indole-3-acetic acid (IAA), 100 mM 1-naphthaleneacetic acid (NAA), 100 mM abscisic acid (ABA) or 100 mM gibberellin (GA) for 3 h under the same growth conditions as described earlier. The 1/2 MS liquid solution without any hormones was used as a control. Another group of seedlings was treated at 4 °C for 1 h and 25 °C was used as a control. Each treatment consisted of three biological replicates.

### 2.8. RNA Extraction and qRT-PCR System

Roots were collected from plants treated with different hormones, and young leaves (the second node from the top) were collected from plants subject to 4 °C. Total RNA was extracted using the RNeasy Plant Mini Kit (Cwbio, Beijing, China). The first-strand cDNA was prepared according to the PrimeScript RT reagent Kit with gDNA Eraser (Cwbio, Beijing, China) protocol. To identify the relative expression level of different *LecRLK* genes under different treatments, qRT-PCR was conducted using FastStart Essential DNA Green Master (Roche, Mannheim, Germany). *CsActin3* (*Csa6G484600.1*) was used as an internal control. qRT-PCR was performed in a total volume of 20 μL, containing 2 μL of cDNA, 10 μL UltraSYBR mixure (Cwbio, Beijing, China), 2 μL gene-specific primers (10 μM) and 6 μL ddH_2_O, using the CFX Connection Real-Time System (Bio-Rad, Hercules, CA, USA) with 40 cycles of 5 s at 95 °C, and 30 s at 60 °C. Each experiment was repeated three times, and each experiment included three biological repeats.

### 2.9. Primer Design and Data Analysis of qRT-PCR

Geneious software (version 2019.0.3) was used to design primers according to the cDNA sequences (Appendix A). PCR-amplified product lengths were about 150 bp. The data from real-time PCR amplification were first analyzed using the 2^−^^ΔΔCT^ method [32]. We then divided the result of each gene under different treatments by the result of the control group and obtained the fold change. Statistical differences were determined by a *t*-test (*p* < 0.05) using Microsoft Excel 2010. The heatmap of the fold-change values of *CsLecRLK*s under five treatments was drawn using the R program (3.5.2).

## 3. Results

### 3.1. Genome-Wide Identification of LecRLKs in Cucumber

We identified a total of 46 *LecRLK* genes, named *CsLecRLK*s, in the cucumber genome through Pfam and SMART search, their coding sequence (CDS) and protein sequences are listed in Appendix A. The total number of *LecRLK*s in cucumber is less than that in *Arabidopsis* (75 *LecRLK* genes) or rice (173 *LecRLK* genes) [4]. The 46 CsLecRLKs were classified into 23 G-type and 22 L-type genes, and one C-type gene based on their extracellular lectin domain. The molecular weights (MWs) of the proteins ranged from 62.5 (Csa1G056960) to 94.5 kDa (Csa3G733860), the isoelectric points (Ips) ranged from 4.98 (Csa4G289630) to 9.51 (Csa1G056960) and the range of CDS length was 1803 to 2502 bp. With the predicted protein structures, it could be considered that most of the CsLecRLKs were localized on the plasma membrane, only Csa7G045520 was located on the extracellular. More information for *CsLecRLK*s, including the length of the gene, the length of CDS, the length of the protein sequence, the protein Molecular weight (MW) and isoelectric point (pI) is listed in Appendix A.

By analyzing the molecular weight of all 46 CsLecRLKs, we found that the weight of G-type CsLecRLKs (83.2 kDa) is generally more than L-type (62.5 kDa) and C-type (74.6 kDa) CsLecRLKs. This may be mainly due to the fact that in addition to the lectin domain, G-type CsLecRLKs often contain EGF and PAN domains (Figure 1). Signal peptides and transmembrane domains (TMs) are critical for protein localization. The software prediction indicated that not all CsLecRLKs had signal peptides and unique TM domains. The loss of a signal peptide or TM domain directly affected the localization of proteins in cells (Appendix A). The plasma membrane localization of most of the CsLecRLKs indicated that they are signal receptors that can sense extracellular signals and then transmit the signals to the interior of the cells.

### 3.2. Phylogenetic Analysis of CsLecRLKs

We constructed a rooted phylogenetic tree using the MEGAX v.10.1.8 (Figure 2). As expected, the phylogenetic tree showed that the CsLecRLK family could be classified into three groups of L-type, G-type and C-type. This result was consistent with the domain-based classification of the CsLecRLK family. The phylogenetic tree indicated that the L-type and C-type groups had a closer relationship. This result was different from previous reports on *Arabidopsis* and rice, which revealed a closer genetic relationship between G-type and L-type groups [4]. As shown in Figure 2, the phylogram of L-type CsLecRLKs could be divided into four sub-groups respectively, by high bootstrap values. However, G-type CsLecRLKs could be divided into at least eight subgroups by bootstrap values. The main reason for this is that there was a big difference in the N-terminals of G-type CsLecRLKs. Some G-type CsLecRLKs contained an S-domain, EGF domain and PAN domain, while others contained only a bulb lectin domain. At the same time, the three LecRLKs from *Arabidopsis* were not isolated, but were incorporated into three different types of CsLecRLK, indicating that different types of LecRLK have high specificity among each other.

### 3.3. Exon–Intron Structural Analysis of CsLecRLKs

The genomic sequence and corresponding cDNA sequence of the *CsLecRLK*s were submitted together to the Gene Structure Display Server (GSDS) together for analyzing their gene structure (Figure 3). The genome sequence lengths of *CsLecRLK*s ranged from 1803 to 6481 bp, and the lengths of CDS ranged from 1674 to 2502 bp. The number of exons of these genes varied from one to nine. Similar to the structure of the *LecRLK* family in other plant genomes [4,15,16], cucumber *C**sLecRLK**s* generally lack introns. Eighty percent of the studied *CsLecRLK*s had less than three exons. Except for *Cs4G296230*, which contained three exons, all L-type *CsLecRLK*s contained only one or two exons, and the C-type *CsLecRLK* (*Csa1G056960*) contained four exons. The G-type *CsLecRLK*s contained one to nine exons. In this group, *Csa7G446780* contains nine exons, the most exons of all *CsLecRLK*s analyzed.

### 3.4. Protein Domain and Motif Analysis of CsLecRLKs

Through the SMART program prediction, we investigated conserved domains that are present in CsLecRLKs. C-type and L-type CsLecRLKs both only contained three base category domains, Lectin domain, transmembrane and kinase domain. However, some G-type CsLecRLKs also contained two other categories of domains, PAN and EGF domains. Among G-type CsLecRLKs, 10 proteins contained PAN and EGF domains at the same time, 5 proteins only contained PAN domain, 8 proteins only contained the EGF domains, and only one contained neither the PAN domain nor the EGF domain. Our result indicates that signal peptides are not necessary in CsLecRLKs. There were 25 CsLecRLKs identified without signal peptides and eight CsLecRLKs with more than two transmembrane domains.

Ten conserved motifs were identified in CsLecRLKs using the MEME program (Figure 3). These motifs were labelled Motif 1 to Motif 10 from the N- to the C-terminus. The details of the conserved motifs are shown in Appendix A. The lengths of these motifs ranged from 15 to 60 residues. Generally, the CsLecRLKs contained 4 to 10 motifs. None of the motifs appeared in all gene family members. Except for Motif 8 and Motif 9, which were only present in the G-type CsLecRLKs, other motifs were present in the three types of CsLecRLKs. With the CDD program (https://www.ncbi.nlm.nih.gov/cdd/), we found that six of these motifs represented different kinase domains (Appendix A), indicating that there may be multiple catalytic phosphorylation sites or targets in each of the CsLecRLKs.

### 3.5. Chromosomal Location and Gene Duplication of CsLecRLKs

We extracted the location data of *CsLecRLKs* and the length data of each chromosome from the cucumber genome annotation files with a series of Perl scripts, and constructed a gene location map using MapChart software. As shown in Figure 4, the *CsLecRLK*s were unevenly distributed across 7 cucumber chromosomes, and genes from the same subfamily on the same chromosome had a tendency to cluster. The number of *CsLecRLK*s on each chromosome varied from two to seven, chromosome 3 contains the largest number of 12 *CsLecRLK*s and chromosome 2 had only two *CsLecRLK*s.

During biological evolution, the generation of a gene family may be caused by tandem duplication and segmental duplication [27,33]. In order to explore whether the *CsLecRLK* gene family also had an expansion caused by the two kinds of duplication, we analyzed the duplication events of *CsLecRLK* genes. The result indicated that although many *CsLecRLK* genes were clustered on the chromosomes, only *Csa1G071170* and *Csa1G071160* were a pair of tandem duplicated genes, their divergence was about 38.61 million years ago (MYA). The other two pairs of duplication events, *Csa1G073890* and *Csa7G048050*, and *Csa3G734030* and *Csa4G296230*, may be have been caused by duplication or ectopia of chromosome fragments during the evolution. These duplicated genes were not in the same chromosome. Their divergence times were 30.96 and 32.35 MYA, respectively. Based on the above results, it could be inferred that duplication events contributed to the expansion of the *CsLecRLK* gene family.

### 3.6. Cis-Acting Elements Analysis on Promoters of CsLecRLKs

Different genes have their own specific *cis*-acting elements on their promoters. Transcription factors can bind to the *cis*-acting elements to regulate the gene expression. Different *cis*-acting elements may respond to different biotic or abiotic stress signals which could induce or inhibit the genes’ expression. Therefore, the *cis*-acting elements analysis of *CsLecRLK* promoters will help us to further understand the function of these genes. We used the Plantcare website (http://bioinformatics.psb.ugent.be/webtools/plantcare/html/) to analyze the promoters of the 1500 bp upstream sequence from the translation initiation site of *CsLecRLKs*, and found that there were 54 typical and functional *cis*-acting elements (Appendix A), which could be divided into four types: light response, stress resistance, plant hormone and others. Among them, 24 *cis*-acting elements were related to light response, 11 were related to hormones including salicylic acid (SA), jasmonic acid (JA), ethylene (ET), gibberellin (GA) and auxin, and 9 were abiotic stress elements. These results suggested that the *CsLecRLK* gene family may be mainly involved in the biological pathway of stress resistance in cucumber. There were six developmentally related *cis*-acting elements, five of which were related to seed development, suggesting that this gene family may play a role in seed development. More details are shown in Appendix A.

### 3.7. Expression Pattern Analysis of CsLecRLK Genes

As a first attempt to provide insights into their potential functions, we used RNA-seq data from 10 tissues of cucumber to investigate the expression of each *CsLecRLK* gene (Appendix A). Most *CsLecRLK*s were expressed at a low level, and some (*Csa6G338050*, *Csa1G071160* and *Csa3G115090*) were barely expressed in any tissue.

RNA-seq analysis showed that the expression of *CsLecRLK*s was ubiquitous. Each tissue contained at least 25 constitutive expression pattern (FPKM ≥ 1) genes (Figure 5A), except for male flower and unfertilized ovary. The expression pattern of all *CsLecRLK*s could be divided into 3 groups based on their expression level in each tissue (Figure 5A). From Groups 1 to 3, the range and level of gene expression decreased successively. Group 1 contained seven genes, which had a high expression level in each tissue with an average FPKM of 18.01. There were 12 genes belonging to Group 2, and they had an intermediate expression level in each tissue, with an average FPKM of 6.48. Group 3 included 27 genes expressed at a low level in each tissue with an average FPKM of 1.29. Except for the C-type *CsLecRLK* (*Csa1G056960*), which belonged to Group 1, the L-type *CsLecRLK*s had higher expression levels than the G-type.

Thirty-six *CsLecRLK*s were expressed in all tissues (FPKM > 0), and 17 genes were constitutively expressed (FPKM ≥ 1). Then, we focused on those genes with relatively high expression (FPKM ≥ 2) and selected five tissues of cucumbers for cluster analysis (Figure 5B), including root, hypocotyl, cotyledon, true leaf and tendril. We found a total of 16 genes that were expressed in all these tissues. Especially, two genes were expressed only in the roots (*Csa2G439210* and *Csa3G115060*), one gene (*Csa3G099580*) was expressed in the tendril, and two genes were only expressed in the cotyledon (*Csa7G446780* and *Csa7G067410*).

### 3.8. Expression Analysis of CsLecRLK Genes in Response to Different Treatments

Gene expression is not only spatiotemporally specific but also can be induced or repressed by hormones and stress. Because most of LecRLKs are receptor proteins on the cell membrane, they can usually sense stimuli at the first time and send signals to intracellular receptors. To uncover all the divergence information for *CsLecRLK*s in different environments over a short period of time, the expression patterns under different hormone treatments, including IAA, GA, ABA, NAA and cold stress treatments, were analyzed by qRT-PCR. The result showed that most *CsLecRLK*s (31/46) responded to at least one treatment (Fold-change > 1 compare to the control group; Significance *p* ≤ 0.05). Overall, there were 20 upregulated events and 38 downregulated events in total (Significance *p* ≤ 0.05). In order to show the experimental results more conveniently and intuitively, the fold-change under different treatments is displayed in the heatmap (Figure 6) based on the data of the qRT-PCR. Firstly, some *CsLecRLK*s (7/46) could be induced or repressed by multiple treatments (treatment number > 3). For instance, *Csa1G071170* could be induced by GA, IAA, NAA and ABA treatments, and *Csa4G005510* was repressed by all treatments except ABA. Secondly, the expression of different *CsLecRLK*s could be induced by different treatments. The cold stress induced or repressed some *CsLecRLK*s genes’ expression. The expression level of four genes changed, and they were downregulated. In contrast, NAA induced or repressed the most *CsLecRLK* genes’ expression. There were 20 genes that responded to NAA treatment, 6 genes were upregulated, and 14 genes were downregulated. A total of 16 *CsLecRLK*s changed their expression level under ABA treatment: 8 genes were upregulated, and 8 genes were downregulated. IAA and GA caused expression level change in nine and eight genes, respectively. The IAA treatment caused one gene’s expression to be upregulated, and those of eight genes to be downregulated. The GA treatment caused five genes’ expressions to be upregulated and those of three to be downregulated. Thirdly, 14 *CsLecRLK*s had different expression patterns under various treatments; for example, *Csa3G734030* could be induced by NAA, and repressed by ABA, while *Csa1G071150* was upregulated under ABA treatment, and downregulated under NAA and IAA treatments. The results indicated that the *CsLecRLK*s have their own response characteristics to hormones and stresses and may play an important role in sensing external stimulus signals. For example, although *Csa1G071160* and *Csa3G115090* were not expressed in the root, our experiment showed that they can be induced by NAA and ABA, respectively. There were 15 genes that did not have significant expression change under different treatments, they were: *Csa1G056960*, *Csa7G029930*, *Csa5G550210*, *Csa4G296250*, *Csa7G048050*, *Csa1G073890*, *Csa4G289620*, *Csa3G115060*, *Csa3G099580*, *Csa1G071270*, *Csa6G516770*, *Csa2G439150*, *Csa1G605730*, *Csa6G338050* and *Csa5G648630*.

## 4. Discussion

The number of *LecRLK* gene family members varies in different plants. In cucumber, 46 *LecRLK* genes were identified and classified into three groups according to the analysis of phylogenetic relationship. The result was consistent with the classification based on the N-terminal domain differences of the three subfamilies. Compared with other plants, there were fewer members of the *LecRLK* gene family identified in cucumber. This phenomenon may be due to the following two reasons. Firstly, the genome sizes are diverse among various plants and may affect the number of gene family members. For example, soybean and wheat have more than 1 and 14 GB genomes and contain 52,051 and 39,238 protein-coding genes, respectively [34,35]. Correspondingly, soybean and wheat contain 189 and 263 *LecRLK* genes, respectively [14,16]. Secondly, compared with other plants, there were fewer duplication events in the *CsLecRLK* gene family in cucumber. Previous studies revealed that *Arabidopsis* experienced two whole-genome duplications (WGDs), while *Populus* experienced only one [36,37]. Due to cucumber lacking a recent WGD [38], there have been fewer opportunities for gene families to expand in the form of duplications during evolution in cucumber. In this study, only three pairs of duplicated genes were identified in cucumber. Although the genome size of *Arabidopsis* (125 Mb) is smaller than that of in cucumber (367 Mb) [38], *Arabidopsis* (75 *LecRLK* genes) has more *LecRLK*s than cucumber (46 *LecRLK* genes). The underlying reason may be the WGD that occurred in *Arabidopsis*. A total of 10 motifs were identified in 46 *CsLecRLK*s, 6 of them were located at the C-terminal of CsLecRLKs and were related to the catalytic activity of kinases (Figure 3 and Appendix A). The result suggested that although the *CsLecRLK* gene family was divided into three subfamilies according to the N-terminal lectin domain, there may be significant differences in target and kinase activity between members in the same subfamily. At the same time, the C-terminus of each CsLecRLK contains at least three motifs related to the catalytic function of kinase. This also implies that each CsLecRLK has more than one site for catalysis or protein interaction. These results indicated that despite the smaller number of *CsLecRLK* gene families, they may participate in multiple biological pathways, making up for the shortfall to some extent.

Another interesting finding in this study was that most *CsLecRLK* genes contained relatively few introns. The gene structure analysis showed that the average intron number per *CsLecRLK*s was 1.5, which was significantly less than the average intron number (4.39) of cucumber genes [38]. This phenomenon was similar to that observed in previous studies in other plants. For example, most members of the *LecRLK* gene family in soybean only have one intron or even none at all [16]. A previous study on *Arabidopsis* and rice also indicated that few genes have introns in the *LecRLK* gene family. For example, only five and eight genes have introns in *Arabidopsis* (75 *LecRLK* genes) and rice (173 *LecRLK* genes), respectively [4]. The lack of introns may be due to these genes acting as the signal receptors in the plant. The compact gene structure is thought to improve the efficiency of gene expression by reducing variable splicing and saving energy, especially those genes needed to respond to various environmental stimuli [39]. There were six *CsLecRLK*s that had more introns than the average intron number of cucumber genes, and four of these did not respond to any hormone or cold stimuli in our study. In the cold stress experiment, although the treatment time was only one hour, it still caused significant changes in the expression levels of four genes. Among these genes, *Csa4G289630*, with the smallest change in expression level, contains two introns, while the other three genes only contained one intron. This evidence further suggests that fewer introns may help *LecRLK*s respond to stimuli quickly.

RNA-seq analysis showed that the expression of *CsLecRLK*s is ubiquitous and their expression level in cotyledons, roots, tendrils and hypocotyls were relatively higher than that in other tissues (Figure 6). Cluster analysis showed that the expression of some *CsLecRLK* genes was tissue-specific. *Csa2G439210* was only expressed in roots and cotyledons and its expression level in roots was 5 times higher than that in cotyledons. *LECRK-IX.1*, the homologous gene of *Csa2G439210* in *Arabidopsis*, was also specifically expressed in roots and guard cells [40]. *Csa5G648630* was expressed in tendrils and cotyledons, but hardly expressed in other tissues. Its homologous gene *LECRK-VI.1* is mainly expressed in cotyledons and guard cells in *Arabidopsis* [41]. *Csa3G099580* is mainly expressed in cotyledons of cucumber, its homologous gene *ARK2* is specifically expressed in cotyledons and vascular bundles in *Arabidopsis.* In addition to responding to pathogen infection, *ARK2* can regulate the development of lateral roots by transferring auxin in a phosphate-starved environment [42]. In this study, the expression level of *Csa3G099580* decreased 1.57-fold after IAA treatment. These results indicated that *LecRLK* genes may have a conserved spatiotemporal expression pattern across different species.

The *LecRLK* family is a special family in the plant genome; to date, this family has not been found in fungi or humans [43]. As the signal station of plants, a unique characteristic of the *LecRLK* family may be closely related to its function of sensing the external environment. Previous studies indicated that some *LecRLK*s are involved in sensing invasion of microorganisms. *SD1-29*, a G-type *LecRLK* in *Arabidopsis*, can identify lipopolysaccharides, which are a secretions from *Gram**-negative Pseudomonas* and *Xanthomonas*
*bacteria* [44]. Some *LecRLK*s will change their expression pattern with the changes of hormone and nutritional conditions. For example, *SIT1*, an L-type *LecRLK* in rice, can mediate salt sensitivity. With increasing NaCl concentration, *SIT1* is activated rapidly, which reduces the survival of rice [45]. In addition, some *LecRLK*s could affect the development and growth of plants; for instance, two L-type *LecRLK*s, *LecRK-IX.1* and *LecRK-IX.2*, induce cell death, thereby increasing plant survival when infected with *Phytophthora* [40]. Analysis of hormone and cold treatments showed that most *CsLecRLK*s (31/46) responded to at least one treatment (Fold-change > 1). We also found a large number of *cis*-acting elements closely related to stress and stimulus response that existed in promoter regions of *CsLecRLK* genes (Appendix A). These results indicate that the *CsLecRLK* gene family may play a key role in responding to environmental stimuli. For example, there were three *cis*-acting elements related to abscisic acid (ABA) response in the promoter of *Csa4G296230* (G-type *CsLecRLK*). Its expression level was upregulated 88-fold in a short period after ABA treatment. The promoter of *Csa5G550210* (L-type *CsLecRLK*) also contains a regulatory element related to abscisic acid response. After ABA treatment, its expression was downregulated 17-fold. An auxin-related regulatory element exists in the promoter of *Csa4G289650*, and expression level of *Csa4G289650* was downregulated 5-fold under IAA treatment. Certainly, although the expression of some *CsLecRLK*s was not detected in tested tissues or not induced by hormones and stress treatments in this study, it may be induced by other hormones or stress, such as salicylic acid, ethylene, salt stress, insect stress and so on. These induction methods and gene function analyses of response to hormones or stress will be the research direction in the future.

## 5. Conclusions

In this study, a total of 46 *CsLecRLK*s were identified using bioinformatics methods, and most of them lacked introns. Forty-six CsLecRLKs were clustered phylogenetically into three distinct subfamilies, and C-type CsLecRLKs and L-type CsLecRLKs had a close relationship. Through the *cis-*elements analysis, many regulatory elements associated with phytohormones and stress were found on the *CsLecRLK*s’ promoters. Transcriptome data demonstrated distinct expression patterns of *CsLecRLK* genes in various tissues. In different tissues, most *CsLecRLK*s were expressed at a low level, and three *CsLecRLK*s were barely expressed in any tissue. In addition, the quantitative real-time PCR (qRT-PCR) analysis showed that each member of the *CsLecRLK* gene family had its own unique expression pattern under hormone and stress treatment. The information obtained in the current study could help to provide a clear understanding of the features of the *LecRLK* gene family in cucumber.

## Figures and Tables

**Figure 1 genes-11-01032-f001:**
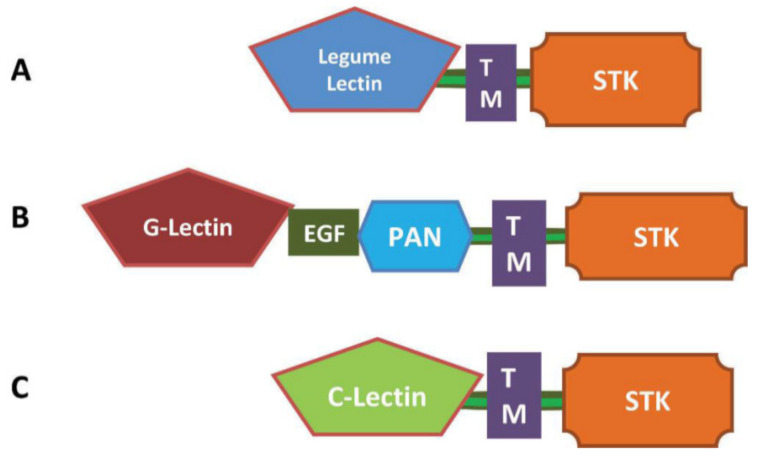
Line Model of the three Lectin receptor-like kinases (LecRLKs) [4]. (**A**) L-type LecRLKs; (**B**) G-type LecRLKs; (**C**) C-type LecRLKs. TM: transmembrane region; STK: cytoplasmic Serine/Threonine kinase domain. PAN: Plasminogen Apple Nematode domain; EGF: Epidermal Growth Factor domain.

**Figure 2 genes-11-01032-f002:**
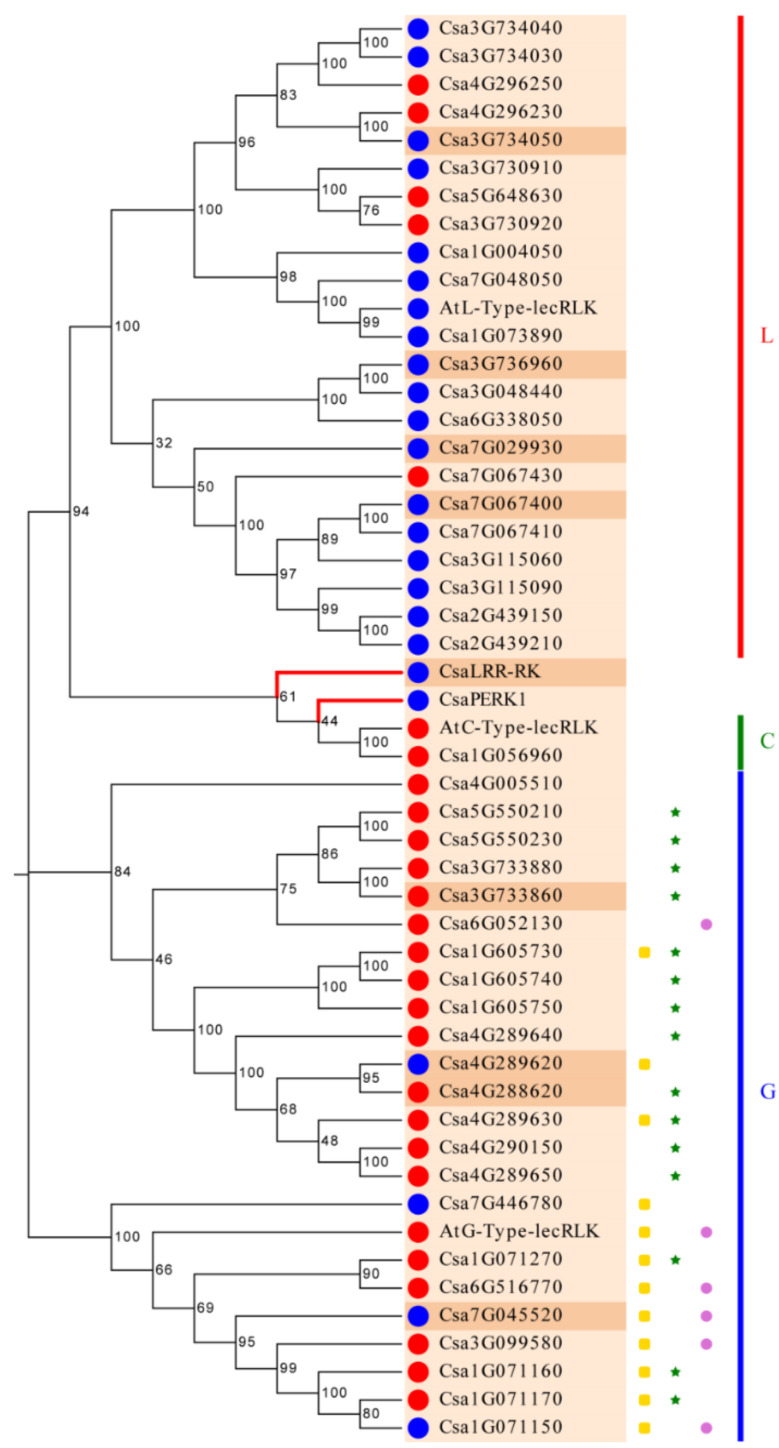
Rooted Phylogenetic tree construction and structure analysis of *CsLecRLK*s. The L, G and C respectively represent the L-type, G-type and C-type LecRLK subfamilies. The blue circles in front of CsLecRLKs represents proteins with a signal peptide, the red circles in front of CsLecRLKs represented protein without a signal peptide. The green stars behind CsLecRLK represents proteins containing Epidermal Growth Factor (EGF) domains, the yellow squares behind CsLecRLKs represents proteins containing Plasminogen Apple Nematode (PAN) domains. The purple circles behind CsLecRLK represents proteins containing S domains. The sandy-brown leaves represents proteins only containing one transmembrane domain, the peach-puff leaves represents proteins containing more than one transmembrane domain.

**Figure 3 genes-11-01032-f003:**
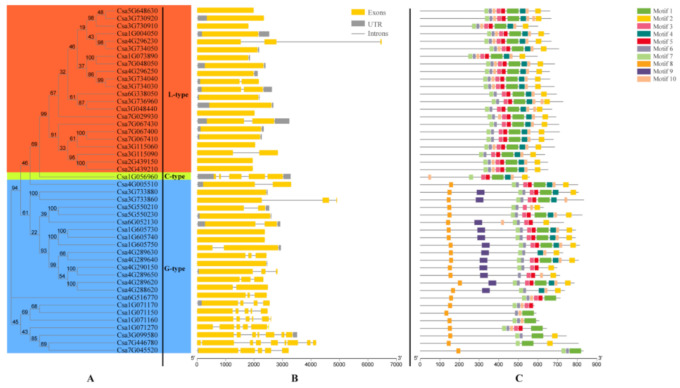
An analytical view of the *CsLecRLK* genes. (**A**) Protein maximum likelihood tree. The tree was constructed using a maximum-likelihood method, and bootstrap values were calculated with 1000 replications using MEGAX; (**B**) Gene structure: The lines represents the introns. The gray squares represents the 3’ UTR and the 5’ UTR. The yellow square represents the exon; (**C**) Protein structure. The search for 10 common motifs shared among the CsLecRLK proteins. Different colors represents different motifs.

**Figure 4 genes-11-01032-f004:**
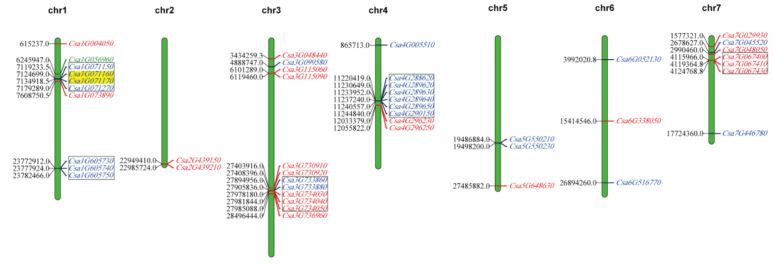
Chromosomal distribution and gene duplications of *CsLecRLK* genes. The number on the chromosome (left) represents the position of *CsLecRLK* genes, and the information on the chromosome (right) represents the gene ID of the *CsLecRLK*s. The tandem duplicated genes are represented by a yellow background, and the gene clusters are boxed together by black lines. The red represents L-type *CsLecRLK*s, the green represents C-type *CsLecRLK*s and the blue represents G-type *CsLecRLK*s.

**Figure 5 genes-11-01032-f005:**
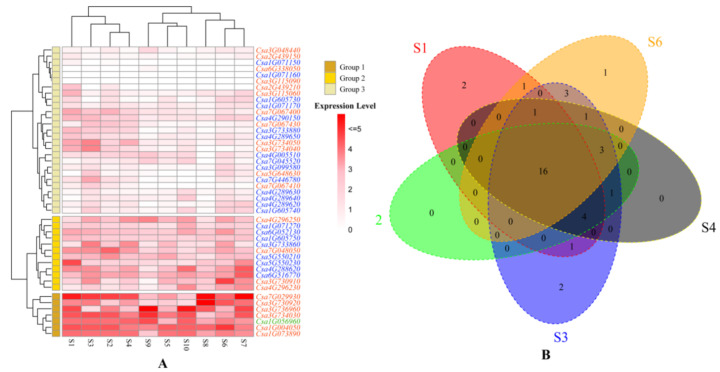
Expression profile of *CsLecRLK* gene family in different tissues. (**A**) Heatmap depicting the expression profile of the *CsLecRLK* family in different developmental tissues. The red represents L-type *CsLecRLK*s, the green represents C-type *CsLecRLK*s and the blue represents G-type *CsLecRLK*s. Genes highly expressed in tissues are colored red, and genes not expressed in tissues are colored blue; (**B**) Venn diagram depicting the distribution of shared expression of *CsLecRLK*s in five typical tissues. The following abbreviations are used: S1: roots of 4 week old seedlings; S2: hypocotyl of 4 week old seedlings; S3: cotyledon of 4 week old seedlings; S4: true leaf of 4 week old; S5: Stem; S6, Tendril; S7: female flower; S8: male flower; S9: unfertilized ovary; S10: peel of unfertilized ovary.

**Figure 6 genes-11-01032-f006:**
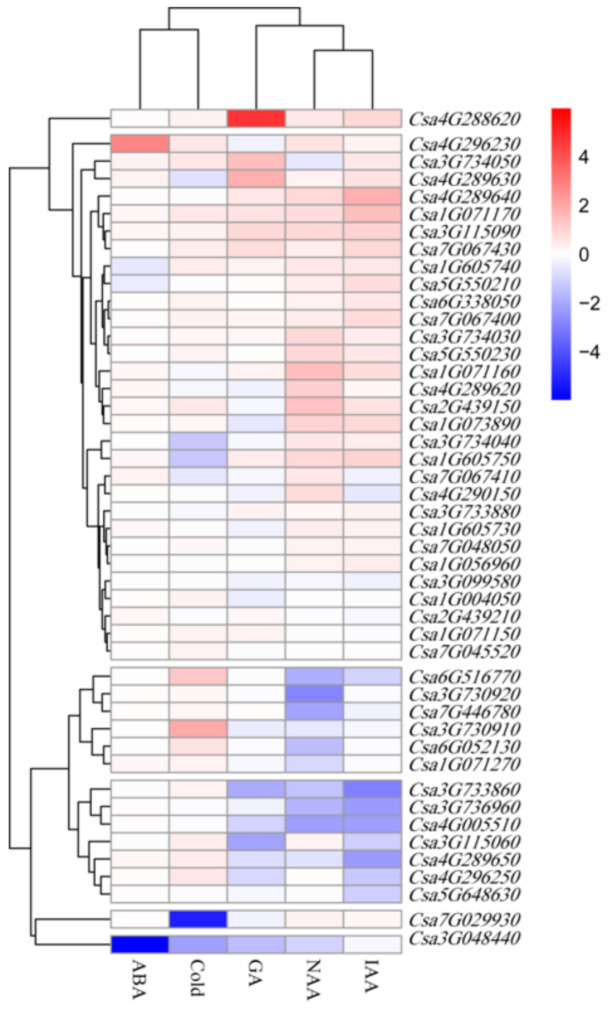
Expression analysis of *CsLecRL*K genes in response to different treatments. The heatmap represents the fold-change of expression level under five treatments: gibberellin (GA), abscisic acid (ABA), 1-Naphthaleneacetic acid (NAA), indole-3-acetic acid (IAA) and cold stress.

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
