# Peer review of "Genome-Wide Identification and Characterization of Lectin Receptor-Like Kinase Gene Family in Cucumber and Expression Profiling Analysis under Different Treatments"

_genes, 2020, doi:10.3390/genes11091032_

Round 1

Reviewer 1 Report

You are necessary to upload file of Supplementary Materials.

Latest MEGA is version X.  Why do you use MEGA 7.0.21.

Table 1 extends the line of the partition.

I think that references No. 40 should write all authors by the name not an initial.

Author Response

Dear reviewer:

    First of all, Thanks for you reviewing my manuscript and giving professional comments. This is my cover letter, I will provide a point-by-point response to your comments.Because we rewrote some parts of MS and sought English Editing Services, the line numbers and reference numbers have been changed.

  1. You are necessary to upload file of Supplementary Materials.

    I remember that I had uploaded the file of Supplementary Materials. It may be unsuccessful because of my unfamiliarity with the operating system. I am very sorry for that. I have uploaded supplementary files to the editor. 

  1. Latest MEGA is version X. Why do you use MEGA 7.0.21.

    Due to the impact of COVID-19, I was unable to return to school to use the Internet for a long time. My home network was not stable, so I could not update the new version of the software in time. Following your suggestion, I have downloaded MEGA X and reanalyzed the data. I have modified the software version in 2.2 of MS. At the same time, I also updated Figure.3.

  1. Table 1 extends the line of the partition.

    Table1 is really too big for MS, so that we are not convenient to edit its position in the text, therefore, we decided to remove it and put it in the file of Supplementary Materials Table S4.

  1. I think that references No. 40 should write all authors by the name not an initial.

    I have rewrited author’ name in MS. Here is the result of the rewrite:

  1. G A, Tuskan.; S, DiFazio.; S, Jansson.; J, Bohlmann.; I, Grigoriev.; U, Hellsten.; N, Putnam; S, Ralph.; S, Rombauts.; A, Salamov., et al. The genome of black cottonwood, Populus trichocarpa (Torr. & Gray). Science (New York, N.Y.) 2006, 313, doi:10.1126/science.1128691.

Reviewer 2 Report

The authors present the results of a bioinformatic analysis of the genome of cucumber to identify most of the LecRLKs. They identified 46 LecRLKs which cluster in the three subgroups G-type, L-Type, and C-type. 

The manuscript is well written in some parts but lacks some details in other. Moreover, some of the statements the authors make in there discussion are hard to follow or not even supported by the presented results. A big concern of mine in the introduction. Some parts of the verbiage in the introduction are very close to the publication of Liu et al 2018 (Liu, P. L., Huang, Y., Shi, P. H., Yu, M., Xie, J. B., & Xie, L. (2018). Duplication and diversification of lectin receptor-like kinases (LecRLK) genes in soybean. Scientific reports8(1), 1-14.). Therefore, I would recomment the authors to rewrite the introduction to avoid potential plagiarism.

Line 32: Plants are not “creatures”

Lines 50-51: I think in the abbreviations got mixed up in this sentence

Lines 58-59: The statement

Lines 68-77: I am not sure if the statement of a “complete” identification and analysis of the entire LecRLK gene family in cucumber is appropriate here, especially since the authors mention in the following sentence that the genes involved in this study are predicted.

Chapters 2.6, 2.7, and 2.8: I don’t understand why the authors used public available data for the expression profiling, especially they grew and analyzed their own samples anyways.

Figure 1: Abbreviations for EGF and PAN are missing

Figure 2: Outgroup is missing. The circular representation is hard to read; therefore, I recommend a linear representation. Bootstrap values are generally low in that figure. The low bootstrap values and missing outgroup makes it hard to believe in the phylogenetic analysis of the authors.

Figure 5: the color code for the groups and CsLecRLK should be different using the same color for two different coding systems is highly confusing.

Lines 385-393: The examples wheat and soybean don’t make much sense here because both crops are self-compatible. Further, I don‘t understand the authors argument mentioned in lines 389 – 393. Why should self-pollinating plants distinguish between pollen sources? Or do the authors mean self-incompatible plants? In that case eucalyptus is a bad example because of its potential of self-pollination.

Lines 426-439: The authors present a lot of previously mentioned results. This abstract should be shortened.

Line 443: Plants are not creatures, they are plants. See definition of creature:

https://www.collinsdictionary.com/dictionary/english/creature

Author Response

Dear Reviewer

    Thanks for your serious and sincere comments after patiently reviewing our manuscript, most of which are caused by my lack of scientific research experience and writing ability. I am very sorry for that. We have revised these mistakes try our best. This is my cover letter, I will provide a point-by-point response to your comments. Please review them and give your valuable suggestions. Because we rewrote some parts of MS and sought English Editing Services, the line numbers and reference numbers have been changed.

  1. A big concern of mine in the introduction. Some parts of the verbiage in the introduction are very close to the publication of Liu et al 2018 (Liu, P. L., Huang, Y., Shi, P. H., Yu, M., Xie, J. B., & Xie, L. (2018). Duplication and diversification of lectin receptor-like kinases (LecRLK) genes in soybean. Scientific reports, 8(1), 1-14.). Therefore, I would recomment the authors to rewrite the introduction to avoid potential plagiarism.

    Thanks for your careful examination of our MS. We realize that we made a mistake, but please believe that we did not deliberately copy the content of others' research. Our previous Introduction is indeed somewhat similar to Publication of Liu et al 2018.We have revised the fourth paragraph of Introduction. We also revised the second paragraph of introduction to make the article more substantial. Please check them. If it still does not meet your expectations, we are willing to make modifications again.

  1. Line 32: Plants are not “creatures”

    Because I am not a native English speaker, I have a deviation in understanding the definition of the ‘creatures’, and I have modified the MS. The revised sentence is: In order to better adapt to the living environment, plants have evolved a complete set of signal receptor proteins during the long-term evolutionary process. In order to avoid similar mistakes happening in MS, we have edited our MS on the professional website.

  1. Lines 50-51: I think in the abbreviations got mixed up in this sentence

    I have adjusted the abbreviations to the right place. The revised sentence is: Plasminogen Apple Nematode (PAN) domain and Epidermal Growth Factor (EGF) domain or one of them.

  1. Lines 58-59: The statement

    I have modified this sentence and added reference in the MS. The revised sentence is: Although C-type lectins are present in large number in mammalian system, C-type LecRLKs seem to be scarce in plants [3]. For example, only one gene encoding for C-type LecRLK exists in rice and Arabidopsis respectively [4].

    [3].  Klaas, B.; Francine, G. Arabidopsis L-type lectin receptor kinases: phylogeny, classification, and expression profiles. J EXP BOT 2009, 15.

    [4].  Vaid, N.; Pandey, P.K.; Tuteja, N. Genome-wide analysis of lectin receptor-like kinase family from Arabidopsis and rice. PLANT MOL BIOL 2012, 80, 365-388.

  1. Lines 68-77: I am not sure if the statement of a “complete” identification and analysis of the entire LecRLK gene family in cucumber is appropriate here, especially since the authors mention in the following sentence that the genes involved in this study are predicted.

    Thanks for your comment. It's true that our statement was not precise enough, so we revised it: In this study, we used bioinformatics methods to identify LecRLK genes from cucumber genome, and analyzed their phylogenetic relationship, gene structure, conserved domain, gene duplications, chromosome distribution and cis-acting elements on the promoters. Furthermore, we profiled the expression of the predicted genes in different tissues and response to gibberellin (GA), abscisic acid (ABA), 1-Naphthaleneacetic acid (NAA), indole-3-acetic acid (IAA) and cold treatments in cucumber. Our study provides valuable information for further functional research on the LecRLK gene family in cucumber.

  1. Chapters 2.6, 2.7, and 2.8: I don’t understand why the authors used public available data for the expression profiling, especially they grew and analyzed their own samples anyways.

    Thanks for your serious comment, consistent experimental materials will make the research more convincing. But I am sorry for that due to time and cost constraints, we did not perform RNA-seq for all tissue by our material, so we used a public data. The purpose of our study is to generally understand the expression pattern of CsLecRLKs in different tissues of cucumber, without deliberately distinguishing materials. In addition, some studies of other gene families in plants also used public databases like us.

    For example, publication of Tao et al 2018(https://doi.org/10.1186/s12864-018-4880-x); publication of Yang et al 2016 (https://doi.org/10.1186/s12864-016-3026-2) and publication of Wang et al 2018 (https://doi.org/10.3390/genes9010054) .

  1. Figure 1: Abbreviations for EGF and PAN are missing

     I have added abbreviations for EGF and PAN in footnotes of Fig.1

    PAN: Plasminogen Apple Nematode domain, EGF: Epidermal Growth Factor domain

  1. Figure 2: Outgroup is missing. The circular representation is hard to read; therefore, I recommend a linear representation. Bootstrap values are generally low in that figure. The low bootstrap values and missing outgroup makes it hard to believe in the phylogenetic analysis of the authors.

    Thanks for your comment. I have updated Figure.2 in MS. In the new Figure.2, The phylogram was rooted using distantly related two RK sequences of cucumber, CsaLRR-RK (Csa3G126190) and CsaPERK1 (Csa2G004170). We also chose At1G52310 (C-type LecRLK of Arabidopsis), At4G04960 (L-type LecRLK of Arabidopsis) and At1G61550 (G-type LecRLK of Arabidopsis) as outgroup of each type LecRLK. We also modified contents of 2.2 and 3.2. Please check them out. If my modification cannot meet you, I am willing to make modification again.

  1. Figure 5: the color code for the groups and CsLecRLK should be different using the same color for two different coding systems is highly confusing.

    Thanks for your suggestion. I have changed color of group.

  1. Lines 385-393: The examples wheat and soybean don’t make much sense here because both crops are self-compatible. Further, I don‘t understand the authors argument mentioned in lines 389 – 393. Why should self-pollinating plants distinguish between pollen sources? Or do the authors mean self-incompatible plants? In that case eucalyptus is a bad example because of its potential of self-pollination.

    Thanks for your comment, our statement is really insufficient to support this argument. So to avoid controversy, we have decided to delete the second reason.

  1. Lines 426-439: The authors present a lot of previously mentioned results. This abstract should be shortened.

    According to your suggestion, I have deleted some contents appropriately, and the revised paragraph is as follows:

    RNA-seq analysis showed that the expression of CsLecRLKs is ubiquitous and their expression level in cotyledons, roots, tendrils and hypocotyls was relatively higher than that in other tissues (Fig. 6). Cluster analysis showed that the expression of some CsLecRLK genes were tissue-specific. The Csa2G439210 was only expressed in roots and cotyledons and its expression level in roots was 5 times higher than that in cotyledons. LECRK-IX.1, the homologous gene of Csa2G439210 in Arabidopsis, was also specifically expressed in roots and guard cells [43]. The Csa5G648630 was expressed in tendril and cotyledon, but hardly expressed in other tissues. Its homologous gene LECRK-VI.1 was mainly expressed in cotyledons and guard cells in Arabidopsis [44]. The Csa3G099580 was mainly expressed in cotyledons of cucumber, its homologous gene ARK2 was specifically expressed in cotyledon and vascular bundle in Arabidopsis. In addition to responding to pathogen infection, ARK2 can regulate the development of lateral roots by transferring auxin in a phosphate-starved environment [45]. In this study, the expression level of Csa3G099580 decreased by 1.57 times after IAA treatment. These results indicated that LecRLK genes may be conservative in spatiotemporal expression pattern in different species.

  1. Line 443: Plants are not creatures, they are plants. See definition of creature.

    Thanks for your reminder. I was not clear about the definition of ‘creature’ when I wrote the MS. Now, I have modified the original sentence to: As the singal station of plants.

Reviewer 3 Report

Dear Authors, you should address my comments inside the text.

Author Response

Thanks you for reviewing my manuscript and patiently pointing out the mistakes in the MS. This is my cover letter, I will provide a point-by-point response to your comments. Because we rewrote some parts of MS and sought English Editing Services, the line numbers and reference numbers have been changed.

  1. We have added space between each reference number and content.
  2. Line 45: We have rewritten stimulus into stimuli.
  3. Line 60: You noted me to remove the "The role of LecRLK...". We have revised this paragraph.
  4. I have rewrite each headlines (2.1- 2.8, 3.1-3.8). They are:

2.1 Identification of LecRLK Genes in Cucumber; 2.2 Phylogenetic Analysis; 2.3 Conserved Domain, Motif Ientification and Gene Structure Analysis; 2.4 Gene Location and Duplication Analysis of CsLecRLKs; 2.5 Analysis of Cis-acting Elements; 2.6 Expression Pattern Analysis; 2.7 Hormones and Cold Treatments; 2.8 RNA Eextraction and qRT-PCR System; 2.9 Primer Design and Data Analysis of qRT-PCR; 3.1 Genome-wide Identification of LecRLKs in Cucumber; 3.2 Phylogenetic Analysis of CsLecRLKs; 3.3 Exon–Intron Structural Analysis of CsLecRLKs; 3.4 Protein Domain and Motif Analysis of CsLecRLKs; 3.5 Chromosomal Location and Gene Duplication of CsLecRLKs; 3.6 Cis-Acting Elements Analysis on PromoterS of CsLecRLKs; 3.7 Expression Pattern Analysis of CsLecRLK Genes; 3.8 Expression Analysis of CsLecRLK Genes in Response to Different Treatments

  1. Line 160-176: According to your suggestion, we split section 2.8 into two paragraphs, producing section 2.9 ‘2.9 Primer Design and Data Analysis of qRT-PCR
  2. Line 281: We have removed a space between ‘many’ and ‘CsLecRLK’
  3. Line 377: I have rewritten ‘from’ into ‘in’.
  4. Line 440: I have removed a space between ‘treatment.’ and ‘These’
  5. Line 478: I have removed ‘that’.
  6. Line 461-462: I have modified the original sentence to ‘An auxin-related regulatory element exsits in the promoter of Csa4G289650, and expression level of Csa4G289650 was down-regulated 5 fold under IAA treatment..
  7. Line 462-466: I have modified the original sentence to ‘Certainly, although the expression of some CsLecRLKs was not detected in tested tissues or not induced by hormones and stress treatments in this study, it may be induced by other hormones or stress, such as salicylic acid, ethylene, salt stress, insect stress, and so on.’

Round 2

Reviewer 1 Report

I checked revised-manuscript and have a following comment.

One minor problem

It has better that you should change from MEGA 7.0.21. to MEGA X in legend of Figure 3.

Author Response

Dear Reviewer 1

    Thanks for your serious and sincere comments, we have modified the mistake in legend of Figure 3. Rewrote MEGA 7.0.21 to MEGA X.

Reviewer 2 Report

The authors improved the manuscript substantially. There are only minor changes which have to made.

Lines 52 - 57: these sentences are a little bit confusing, are G-type lecRLKs historically known as B-type or S-Domain or both? Please rewrite these sentences for a better understanding.

Line 82: the „i“ in immune is missing

Line 236: in the material and method you mention that the phylogenetic tree is rooted but here you say it’s unrooted. Similarly, in the caption of figure 2 you say the Tree is unrooted, but I think you mean rooted.

Lines 236-251: you use the word subgroup in two different contexts. Maybe you could say that „As expected,

the phylogenetic tree showed that the CsLecRLK family could be classified into the three groups of

L-type, G-type, and C-type.“

Author Response

Reviewer 2

    Thanks for your serious and sincere comments, I provide a point-by-point response to your comments. Please check them in attachment.
